# Interactions between Entomopathogenic Fungi and Insects and Prospects with Glycans

**DOI:** 10.3390/jof9050575

**Published:** 2023-05-15

**Authors:** Dongdong Liu, Guy Smagghe, Tong-Xian Liu

**Affiliations:** 1Institute of Entomology, Guizhou University, Guiyang 550025, China; 2Institute of Plant Health and Medicine, Guizhou University, Guiyang 550025, China

**Keywords:** mycopesticides, *Beauveria bassiana*, *Metarhizium anisopliae*, insect, integrated pest management, sustainability, glycans, immunity

## Abstract

Concerns regarding the ecological and health risks posed by synthetic insecticides have instigated the exploration of alternative methods for controlling insects, such as entomopathogenic fungi (EPF) as biocontrol agents. Therefore, this review discusses their use as a potential alternative to chemical insecticides and especially focuses on the two major ones, *Beauveria bassiana* and *Metarhizium anisopliae*, as examples. First, this review exemplifies how *B. bassiana*- and *M. anisopliae*-based biopesticides are used in the world. Then, we discuss the mechanism of action by which EPF interacts with insects, focusing on the penetration of the cuticle and the subsequent death of the host. The interactions between EPF and the insect microbiome, as well as the enhancement of the insect immune response, are also summarized. Finally, this review presents recent research that N-glycans may play a role in eliciting an immune response in insects, resulting in the increased expression of immune-related genes and smaller peritrophic matrix pores, reducing insect midgut permeability. Overall, this paper provides an overview of the EPF in insect control and highlights the latest developments relating to the interaction between fungi and insect immunity.

## 1. Introduction

Insects are incredibly ubiquitous organisms in nature and are found in virtually every corner of the planet. Insect pests can reduce crop yields and, as a result, impact agricultural productivity and horticultural output, posing a formidable threat to food supply and economic stability. Over the past several decades, the employment of chemical pesticides has become increasingly widespread for pest management globally. While these substances offer quick pest eradication, their utilization presents hazards to both human health and the agroecosystem, leading to undesirable effects on natural enemies or pesticide residue [1]. Furthermore, persistent exposure to pesticides has resulted in the development of pest resistance to various chemicals [2,3]. Consequently, concerns regarding the negative effects of chemical insecticides have prompted a focus on eco-friendly and alternative strategies for pest control. Numerous efforts have been made to develop biological control agents (BCAs) as alternatives or supplements to these chemicals. These include the utilization of microbial control agents against insect pests, such as bacteria, viruses and fungi [4,5,6,7,8,9,10]. In 2022, at the Annual Biocontrol Industry Meeting (ABIM) held in Basel, Switzerland, Marrone provided an update on the share of the biological market [11]. The report showed that the fastest-growing segment of the biocontrol market was pest biocontrol, with a compound annual growth rate (CAGR) of 13.6%. Biostimulants and biofertilizers are estimated to have CAGRs of 12.0% and 12.5%, respectively. The biocontrol market is expected to increase from its current value of around $5 billion to $15 billion USD by 2029. Currently, the largest biocontrol category is biochemicals, which include pheromones, plant extracts, and plant growth regulators. However, by 2029, microbial biopesticides are expected to be almost equally significant [11]. In addition, the markets in Latin America and North America are dominated by microbials, with the exception of Europe, where regulatory policies hinder microbial introduction. Direct statistics reveal that biofungicides alone generated $58.8 million in sales during the 2021–2022 harvest season in Brazil [12].

Except for species such as *Lecanicillium* sp., which mostly occurs on phylloplane, entomopathogenic fungi (EPF) are a special group of soil-dwelling microorganisms that possess the ability to infect and kill arthropods through cuticle penetration and proliferation in the hemolymph [13,14]. Some of these fungi are currently used as BCA against insect plant pests and play a vital role in their management, including *Beauveria bassiana* (Cordycipitaceae) Vuillemin and *Metarhizium anisopliae* (Metschnikoff) Sorokin. *Beauveria bassiana* and *M. anisopliae* have shown potential for controlling many economically important insect pests and have been developed as BCAs for agricultural applications (for inundation and inoculation biological control). Table 1 presents an overview of these commercial *B. bassiana*- and *M. anisopliae*-based biopesticides. Since different strains show varied virulence in fields of different regions, we considered the strains and their use against different pest insects in different regions. Additionally, in this review, we utilized *B. bassiana* and *M. anisopliae* as major examples for illustrating the potential of EPF.

There are several reasons why *B. bassiana* and *M. anisopliae* are recognized as the two most known and significant entomopathogens. Apart from their advantages with a broad distribution and host range, which can be used in different agricultural crops and agricultural fields, other characteristics can be shown in terms of their field application and ease of mass production. First, we provide some descriptions of these two entomopathogens. *Metarhizium anisopliae*, which was first described by Metschnikoff in 1879 on infected larvae of the wheat cockchafer *Anisoplia austriaca* (Coleoptera) and later established by Sorokin in 1883 as the green muscardine fungus [15], occurs on a broad range of insect hosts [16,17,18,19,20]. The most comprehensive list of host insects was presented by Veen, who recorded 204 naturally infected insect species from seven orders [21]. Notably, studies by Hussein et al. [22] have reported the prevalence of *M. anisopliae* in cultivated soils. Of all the biopesticides investigated, *M. anisopliae* is the most extensively researched, and its insecticidal properties have been widely investigated in recent studies [23,24,25]. Additionally, its effectiveness in controlling agricultural insect pests has made it a popular BCA alongside the microbial insecticide *Bacillus thuringiensis* Berliner (Bacillales: Bacillaceae) [26]. *B. bassiana*, which was originally isolated from silkworm cadavers by Agostino Bassi in the 19th century, exhibits a broad host range as *M. anisopliae*, infecting more than 200 insect species across six orders and fifteen families [27,28], inhabiting diverse ecosystems such as stored product insects [29,30], bees [31,32], moths [33,34,35,36] and mosquitos [37,38,39] among others [40,41], making it a highly versatile and effective biopesticide.

The key advantage of using EPF as a biopesticide is its specific mode of action, which primarily involves the production of a hypha/penetration peg that can penetrate the insect host, leading to the invasion of the insect’s body and eventual death without further producing toxins that are harmful to non-insect organisms such as mammals or birds. This characteristic makes it a safe and environmentally friendly alternative or supplement to chemical pesticides for controlling insect pests in agricultural settings. The incorporation of EPF with insecticides can potentially reduce the use of chemical insecticides, thereby improving pesticide efficacy and reducing chemical residues and negative side effects in agriculture [42]. Several studies have indicated that the underlying mechanism for this phenomenon could be that insecticides may act as a general stressor by weakening the insect cuticle, reducing the mobility of the target pest due to paralysis, disrupting the removal of fungal conidia via grooming behavior and making the insect more susceptible to the attachment and entry of EPF [43,44]. For now, numerous EPF collections are available, including those maintained by organizations in the USA [45], Europe [46], China [47], and Brazil [48], that could support academic research and field applications. In addition, experiences in mass production have been accumulated; therefore, several mycoinsecticides have been successfully mass-produced for the control of pests [49,50,51] in the formulation of *M. anisopliae* or *B. bassiana*. These fungi can be easily cultured on a large scale and can be formulated into various formulations such as sprays, dust, and granules. Observations of epizootics among insect populations are common, indicating the significant potential of these microorganisms for the regulation of pest species.

Indeed, biological pest control by EPF has immense advantages, but the application of EPF also has some limitations. Firstly, there are still challenges in the isolation and identification of fungal endophytes. Several fungal strains need specific media for their growth and recovery rate, and many fungal strains have been found to be unculturable. Hence, measuring and identifying the endophyte community structure, composition and diversity have been difficult tasks [52]. Secondly, the proper function of EPF needs favorable environmental conditions (a favorable temperature, relative humidity, and pH) for germination and infection in the fields. Additionally, their persistence and infection rate is also highly dependent on changing environmental conditions. Moreover, the mass production of most EPF is costly, which makes the field application of EPF not so cost-effective when compared to the use of chemical insecticides. In addition, the successful application of EPF requires excellent technical expertise for smooth spray coverage before and during its application. Aside from these, there are still other factors that may influence the successful application of EPF, and readers are recommended to refer to the contents in [53,54].

Insects and fungi have been interacting for millions of years; therefore, except for antagonistic relationships, the interactions between these two could also be mutualistic. One case is that fungi could represent a food source for some insects. Studies on species such as ants, termites and some Coleoptera (e.g., the ambrosia beetles and the ship-timber beetles), have shown that these insects could cultivate fungi in their nests as their main food source [55]. As a result of these interactions, many insects have arisen to evolve external cuticular modifications to house fungal symbionts, such as mycangia in beetle-fungus symbioses [56]. Following the paradigm of fungi, host plants and insects in the ecosystem, associated plants are also part of these relationships, and they mediate fungi and insect interactions. Plant-mediated interactions between fungi and insects can also be mutualistic. An extraordinary case of this interaction is that flower organs and nectar are commonly inhabited by yeasts which have a significant impact on the foraging behavior of pollinators and parasitoid attraction. The consumption of nectar colonized by yeasts has been shown to improve bee fitness [57,58,59]. While beneficial microbes could act as plant defense elicitors that confer plant resistance against pests and pathogens. In other aspects, plants could take advantage of fungi to protect themselves from herbivores. One example of this is in the protection of tomato plants against the two-spotted spider mite *Tetranychus urticae* (Acari: Tetranychidae). When the tested fungal strains were applied, *T. urtica*’s survival, egg production and feeding were severely reduced. In this study, all fungal strains studied were shown to negatively affect the spider mite’s performance when applied as a water drench, while arbuscular mycorrhizal fungi-*Rhizoglomus irregularis* (Glomerales) strains were the most promising of all [60]. More examples can be seen in the area of mycorrhiza and endophyte, where “mycorrhiza” describes a type of fungus that has a mutualistic relationship with plant roots, while “endophyte” describes a fungus that lives within above-ground healthy plant tissue and does not seem to harm it. Root-colonizing and endophytic fungi interact with herbivore insects by modulating plant defenses and stimulating the production of plant volatile organic compounds, which attract the natural antagonists of pests [59,61,62,63].

In other aspects, plants can be considered as an active bridge between above- and below-ground organisms, including fungi, insects, and other vertebrate and invertebrate species [64,65]. Plant hormones, including jasmonic acid (JA) and salicylic acid (SA), are the key hormones regulating plant defenses against biotrophic pathogens and insect herbivores with a piercing-sucking feeding mode, such as aphids and whiteflies [66,67]. At present, increasing evidence has shown that the final outcome of plant defenses against various attackers is also dependent on hormones other than JA and SA, such as auxin, ethylene, brassinosteroid, and strigolactone, all of which are important in many aspects of plant growth and development [68,69,70,71,72,73]. The review by Nurmi et al. [74] in this issue provides valid examples of how these hormones influence the interactions between insects and fungi. Additionally, all these interactions hold promise for the application of microbes in the control of pests that are above- and below-ground.

In all, in this review, we present *B. bassiana* and *M. anisopliae* as examples and provide a synopsis of the utilization of EPF for insect control. We explain the fungus’ pathogenicity by delving into the intricacies of the fungus–insect interaction with special emphasis on immune responses. Notably, we also draw attention to a recent discovery, positing that the accumulation of fungal N-glycans may function as a mechanism for the invading fungi to elicit an immune response in insects.

## 2. EPF Pathogenicity and the Interaction between Fungi and Insect

### 2.1. Fungi’s Infection Mode of Action in Insects: Penetration through the Cuticle

On the mode of action, the primary means of entry for most EPFs are via penetration through the host cuticle. A number of steps occur during fungal infection, with the initial adhesion of fungal conidia (asexual spores or fungal seeds) to the host cuticle preceding penetration (Figure 1a). These fungal pathogens are capable of infecting both hard- and soft-bodied insects, as well as a range of other arthropods such as Acari (i.e., ticks and mites) [75]. Among sucking insects, such as aphids and whiteflies, EPF is the primary pathogen as these hosts cannot ingest other pathogens that infect the gut wall [76].

During the course of insect infection, *B. bassiana* and *M. anisopliae* encounter a variety of niches within the host, which are highly variable with respect to the types and abundances of available nutrients [77]. First, the spores adhere to the cuticle of the insects. Additionally, the degree of attachment and the ability of the fungi to penetrate inside the host exoskeleton are crucial determinants for the success rate or extent of infection. Germination represents the second step of the infection process (Figure 1a). The fungus produces an appressorium and hyphal penetration through the cuticle, where primary nutrients, such as the protein and chitin co-polymers, are bound in the cuticle matrix. Indeed, the process of appressorium production is not for all fungal species; an exception can be seen in the example of *Conidiobolus coronatus* (Costantin) Batko (Entomophthorales: Ancylistaceae), which does not produce an appressorium during its infection process. Penetration constitutes the third step, whereby the fungus enters the nutrient-rich insect hemolymph containing accessible sugars, proteins, and lipids. During this process, the fungus spreads within the hemolymph and produces toxins. After this, the mycelium grows on the cadaver and produces new conidia (Figure 1a). The conidia of *B. bassiana* are typically cylindrical or oval in shape, measuring approximately 3–4 microns in length, and are borne on long, slender stalks known as conidiophores. The infected insect may also exhibit deformities and mummification as the disease progresses. These features are critical for accurate diagnosis and the identification of infected insects in the field (Figure 1b). For *M. anisopliae*, infections of insects are easily recognized a few days after death. Initially, the fungal hyphae appear white, but as conidia form and mature, they often take on a characteristic olive-green color (Figure 1c).

**Figure 1 jof-09-00575-f001:**
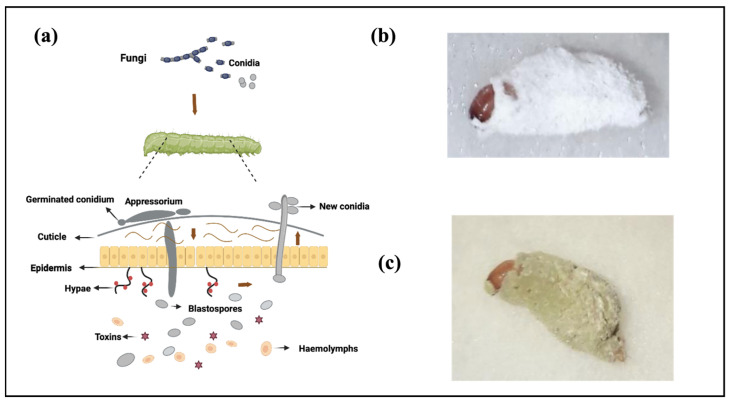
A schematic overview of the infection process of EPF by penetration through the cuticle, and photos of pathogen infection. (**a**) The process of fungal infection on the host cuticle. First, adhesion of conidium to the insect epicuticle; second, spore germination results in appressorium (penetration pegs) penetrating the host cuticle and this follows the hypha/penetration peg entering inside the insect hemocoel; next, the fungal proliferate in the host and produce toxins; to accomplish the infection, the fungal cells differentiate into yeast-like cells called blastospores, and later mycelium grows on cadaver and produces conidia that come out of the insect under suitable environmental conditions and spread to other insects. The arrows indicate the direction of fungal growth. The figure is drawn with BioRender.com. (**b**,**c**) Photo of red palm weevil (*Rhynchophorus ferrugineus*) killed by *B. bassiana* and *M. anisopliae,* respectively. Photos from [78].

### 2.2. Ways of Interaction between Fungi and Insects

#### 2.2.1. Interaction of the Fungus with the Microbiome of the Insect

Insects harbor microbes in various parts of their bodies, including their cuticle surfaces, digestive tract, and tissues or cells. Interactions between insects and their microbes can be classified into two categories: interactions with cuticle surface microbes and interactions with gut microbes. The consequence of insect microbes on the development and survival of EPF can vary depending on the diversity of the insect microbiome, which can differ between developmental stages and in response to environmental conditions. Different studies have shown that both *B. bassiana* and *M. anisopliae* can invade the cuticle and survive within insect cells, potentially coming into direct contact with intracellular endosymbionts [37,38]. Examples of interactions between fungi and insect cuticle microbes are summarized in Boucias’s review [79]. Overall, the cuticle is generally unsuitable for microbes due to its hydrophobic nature, but specialized structures in the exoskeleton may provide habitats for specific microbial taxa. Additionally, for mutualistic relations between fungi and insects, cuticles have evolved to inhabit specific fungi [80,81].

The insect gut has long been targeted for insect control, and research has recently focused on gut microbes and their interactions with EPF. The number and complexity of the gut microbiome may vary in different regions and depend on the insect’s developmental stage and taxa. All insect orders examined to date possess microbial symbionts that contribute to the defense against other microbes. Recent studies have reported that gut microbiota can suppress or promote infections by *B. bassiana*. For example, *B. bassiana* can interact with gut microbiota and accelerate mosquito mortality. Mosquitoes with gut microbiota died significantly faster than those without microbiota after topical fungal infection [82]. Furthermore, the fungal infection caused the dysbiosis of the mosquito gut microbiota, with a significant increase in their gut bacterial load and a decrease in bacterial diversity. The interplay between *B. bassiana* and the gut microbiota of bark beetles (*Dendroctonus valens*) also accelerated mortality, and the gut bacterial community was altered by *B. bassiana* [83]. In a more recent study, Wang et al. [84] found dramatic changes in the gut bacterial community structure in the brown planthopper (*Nilaparvata lugens*) after *M. anisopliae* infection. There was a significant increase in the bacterial load, a decrease in bacterial community evenness, and significant shifts in dominant bacterial abundance at the taxonomic level below the class.

#### 2.2.2. Stimulation of the Insect Immune-Competence by EPF

Fungi can express highly conserved pathogen-associated molecular patterns (PAMPs) that are recognized by pathogen recognition receptors (PRRs) and expressed on host phagocytes. For instance, glucans on the fungal cell surface can serve as PAMPs and be recognized by PRRs expressed on hosts, inducing the humoral and cellular immune responses in insects [85]. However, behavioral avoidance is considered the most effective defense against pathogens. Examples of this include social insects, such as the termite *Macrotermes michaelseni* that can ascertain the virulence of *Metarhizium* and *Beauveria* strains (Table 1) from a distance and is, thus, more strongly repelled by more virulent strains [86]; additionally, the bug *Anthocoris nemorum* avoids foraging and ovipositing on plants contaminated with *Beauveria* spores [87]. 

Before the pathogen encounters the host immune system, the infection begins with the attachment of single-cell dispersive forms of the fungus, e.g., conidia or blastospores, to the insect cuticle, as we discussed in the former section (Figure 1). The insect cuticle itself is a highly heterogeneous structure that can vary greatly in composition even during the various life stages of a particular insect. As summarized in [88], the epicuticle or outermost layer provides a hydrophobic barrier that is rich in lipids and is followed by the procuticle that contains chitin and sclerotized protein, which can typically be divided into the exo-, meso-, and endo-cuticular layers. The procuticle, in turn, is followed by the cells that constitute the epidermis and surrounds the internal structures of the insect. When fungi come into contact with an insect, they use various mechanisms to breach the insect’s cuticle and enter the body cavity [89]. This includes the expression of a variety of hydrolytic enzymes, e.g., proteases, chitinases, and lipases, and other factors, that promote the germination and growth of the fungus across the surface of the host and the subsequent penetration of cuticular layers [90].

Once inside, the fungi proliferate and release enzymes and toxins that break down the insect’s tissues and organs, ultimately leading to the death of the insect. *B. bassiana* synthesizes various secondary metabolites upon the invasion of insect hosts, including beauvericin, bassianin, bassianolide, beauverolides, tenellin, oosporein, and oxalic acid. These toxins facilitate host parasitism and cause mortality. For instance, oosporein, a dibenzoquinone toxin secreted by *B. bassiana* belonging to Sordariomycetes, Cordycipitaceae, represses immune responses in the mosquitoes’ midgut, causing dysbiosis, and subsequently triggers bacterial translocation from the gut into the hemocoel [82]. In the case of *M. anisopliae*, the primary toxic metabolites produced by this species include destruxins (six types) and cytochalasins (C and D), which were identified by Roberts and Hajek [91]. Recent studies have also isolated hydroxyfungerins A and B from *Metarhizium* sp.’s culture broth [92,93]. Additionally, both *B. bassiana* and *M. anisopliae* are believed to employ various extracellular proteases to facilitate pathogenic processes [94,95].

When the fungus breaches the cuticle and penetrates into the host integument by the penetration pegs and/or appressoria, which enable the growing hyphae to penetrate into the host integument, it reaches the insect’s innate immune system. The humoral response of the insect involves the production of antimicrobial peptides (AMPs) in the Toll immune pathway, which is secreted into the body cavity upon infection [96,97,98,99]. For details, the Toll receptor signaling pathway is initiated by the binding of an endogenous peptide ligand termed Spätzle. Spätzle is synthesized as an inactive precursor protein that is cleaved by the protease Easter. Easter is also generated from the precursor protein in a series of protease activation cascades. The binding of the Toll receptor causes recruitment to the membrane of the adapter Tube and the protein kinase Pelle, and finally, this leads to the degradation of the Cactus and the nuclear localization of NF-κB transcription factors Dif and Dorsal. These transcription factors induce the expression of antifungal genes such as Drosomycin (Drs) and Metchnikowin (Mtk) (Figure 2). For Drs, it is a member of the cysteine-stabilized α-helical and β-sheet (CSαβ) superfamily, consisting of 44 amino acid residues with an α-helix and a three-stranded β-sheet. This peptide is fortified by four disulfide bridges and has a limited antimicrobial range, specifically showing significant antifungal properties [100].

Cellular immunity is mediated by hemocytes, including phagocytosis, nodulation and encapsulation. Cellular responses, which involve hemocytes present in the body cavity, are not fully understood compared with the humoral immune response, but they are thought to play a significant role in conjunction with humoral defenses in eliminating insect-invaded bacteria. In *Drosophila*, cellular and humoral responses act together to combat infection [102], with cellular defenses playing a bigger role through nodulation and phagocytosis to eliminate the bacteria (Figure 2). In the later stage of the infection, this bacteria elimination could minimize the infection by the pathogen and facilitate microbe clearance by a later humoral response [103]. For example, in the study with the infection of *B. bassiana* in a yellow peach moth, *Conogethes punctiferalis*, significant decreases in the total and differential hemocyte counts were recorded over time in the larvae after they were injected with *B. bassiana* conidia. Additionally, hemocyte-mediated phagocytosis and nodulation were initiated in the hemolymph of larvae from the *B. bassiana* conidia challenge [104]. In another moth, *Spodoptera frugiperda*, the injection of *Metarhizium rileyi* (Farlow) Samson (Hypocreales: Clavicipitaceae) blastospores decreased the number of *S. frugiperda* hemocytes and impaired host cellular reactions such as nodulation, encapsulation and phagocytosis [105].

In other aspects, the activation of host defenses, in addition to impacting resident microbes, may suppress both the host and the invasive mycopathogen. For example, exposure to a high number of conidia may cause a massive upregulation in the phenoloxidase cascade, leading to the production of toxic quinones that suppress fungal development and/or kill the host [106]. To obtain additional information regarding the relationship between immunity and fungi, the readers are referred to Lu and Leger [107].

## 3. New Interesting Findings on the Interaction of Fungi in Insects

Fungal cell walls are complex and dynamic structures that are essential for fungal growth, development, and survival. They play a critical role in protecting fungal cells from environmental stresses and providing structural support [108,109]. The fungal cell wall is composed of several layers, with the composition and organization of each layer varying depending on the fungal species, morphotype, and growth stage. Fungal cell walls are primarily comprised chitin and β-glucans, which form an inner rigid core (Figure 3) [110]. Chitin is a polymer of N-acetylglucosamine (GlcNAc), while β-glucans are polysaccharides composed of glucose monomers that are linked by β-1,3- or β-1,6-glycosidic bonds. These polysaccharides provide strength and rigidity to the cell wall, making it resistant to mechanical stress [111,112]. In addition to chitin and β-glucans, the outer layer of the fungal cell wall contains various components, including polysaccharides and glycoproteins. In yeast cells, for example, the outer layer of the cell wall is enriched in mannosylated glycoproteins, also known as mannoproteins. These glycoproteins form a top layer on the cell surface and play a critical role in fungal cell adhesion and recognition (Figure 3). Mannoproteins are N-glycosylated or O-glycosylated proteins that are modified with various types of glycans [113,114]. The N-glycosylation pathway in fungi is conserved and produces a wide range of glycans [115]. These glycans can be classified into four main categories based on their structure: pauci-mannose glycans, high-mannose glycans, hybrid glycans, and complex glycans [116,117]. Pauci-mannose glycans are composed of one to three mannose residues and two GlcNAc residues at their base. High-mannose glycans are composed of four or more mannose residues and two GlcNAc residues at their base. Hybrid glycans contain a core structure of two GlcNAc and three mannose residues, with branches consisting of additional sugars such as galactose, sialic acid, and fucose. Finally, complex glycans are branched N-glycans that contain a variety of different sugar residues, including GlcNAc, galactose, and sialic acid. Mannoproteins on the cell surface of yeast cells belong to the high-mannose category, which contains four or more mannose residues and two GlcNAc residues at its base. The exact composition of these glycans can vary depending on the type and number of sugar molecules and the location of the branches (Figure 3).

The presence of N-glycans on the surface of fungal cells is an important factor in fungal biology and pathogenicity [119]. For example, glycoconjugates containing mannose, fucose, or GlcNAc on the surface of some fungal pathogens can be recognized by host mannose receptors on the cell, triggering signaling pathways that are involved in the induction of cytokine production [120,121]. In accordance with this, in one newest research on the glycans and immune response in insects, the authors found that in larvae of the Colorado potato beetle (*Leptinotarsa decemlineata*), after RNAi of the expression of *Mannosidase-Ia* (*ManIa*), which is responsible for the transition from high-mannose to paucimannose glycans [122], the peritrophic matrix pore size width in the *ManIa^RNAi^* insects was decreased by nearly 10 percent when compared to the control *GFP^RNAi^* insects (Figure 4). Interestingly, these smaller pores were connected to the observation of thinner microvilli on the epithelial cells of the midgut of *ManIa^RNAi^* insects, and in addition, these observations in the insect midgut agreed with an accumulation of high-mannose N-glycans in ManIa-silenced insects [123]. The peritrophic matrix is a physical barrier in the gut of insects and functions to protect the midgut epithelium from mechanical damage and harm from pathogens, toxins, and other damaging chemicals [124]. Based on the functions of the peritrophic matrix, the authors brought up the hypothesis that accumulated high-mannose glycans could simulate the cell wall structure of the fungi to trigger an immune response in the insect. They also made the speculation that a decreased pore size could be a protective response to prevent potential pathogens from gaining access to the midgut epithelium. This hypothesis was co-supported by the strong increase in transcription levels of the anti-fungal peptide drosomycin-like in the *ManIa^RNAi^* insects [123], although further research is required to elucidate this possibility. Nonetheless, we believe that such information may lead to novel approaches to improve the efficacy of pest control [125] and could be used for the rational design of strategies to increase the effectiveness of EPF for pest control applications.

## 4. Conclusions

The cosmopolitan existence and rich diversity of EPF contribute significantly to insect population regulation. In the agricultural sector, commercial formulations of EPF have shown efficacy as alternative control agents to chemical pesticides. Among the commercially produced entomopathogens, *B. bassiana* and *M. anisopliae* are the most important bioinsecticides. This review provided examples of *Beauveria-* and *Metarhizium*-based bioinsecticides, information on their infection mechanism and the possible modes of action for these two EPF insects. EPF penetrates their insect hosts through the spore adhesion of the cuticle and germination of the conidia, followed by the formation of an appressorium, and after that, the fungus penetrates the cuticle. Later, the fungus grows/proliferates in the hemocoel and produces blastospore toxins, ultimately leading to insect death. Additionally, we hypothesized that the glycans present on the fungal cell surface could act as motifs that elicit immune responses during the infection process. We also think that cellular responses are important, but we do not know so much about these defense systems. Here, we believe that modern omics-research and RNAi and CRISPR/Cas technologies could help elucidate the function of the many defense genes involved [126,127].

Overall, this review emphasizes the potential of EPF as bioinsecticides and underscores the significance of comprehending the mechanisms underlying their infection processes and immune responses elicited while also highlighting a recent study related to glycosylation and insect biocontrol, which suggests a new way of controlling insect pests by manipulating glycosylation. The study of glycosylation and its role in insect biocontrol represents a promising avenue for future research and has the potential to advance our understanding of fungi.

## Figures and Tables

**Figure 2 jof-09-00575-f002:**
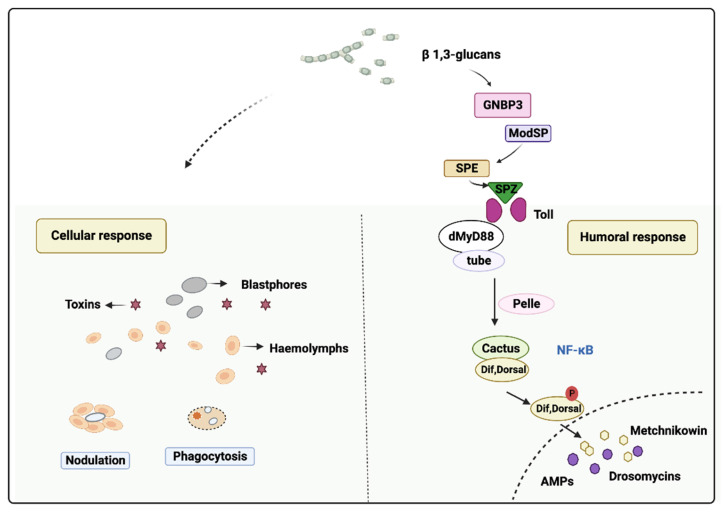
The process of humoral and cellular immune response. The Toll pathway is activated by Gram-positive bacteria and fungi. The β-1,3-glucans of fungi cell wall is recognized by GNBP3, then these interactions initiate protease cascades that converge at the level of the serine protease ModSP, which activates the protease Grass, in turn, activates the Spätzle processing enzyme (SPE). The SPE binding of the Toll receptor causes recruitment to the membrane of the adapter Tube and the protein kinase Pelle, and finally this leads the degradation of the Cactus and nuclear localization of NF-κB transcription factors Dif and Dorsal. The process of cellular response eliminates the pathogens by means of nodulation and phagocytosis. The figure was modified from [101] and drawn with BioRender.com.

**Figure 3 jof-09-00575-f003:**
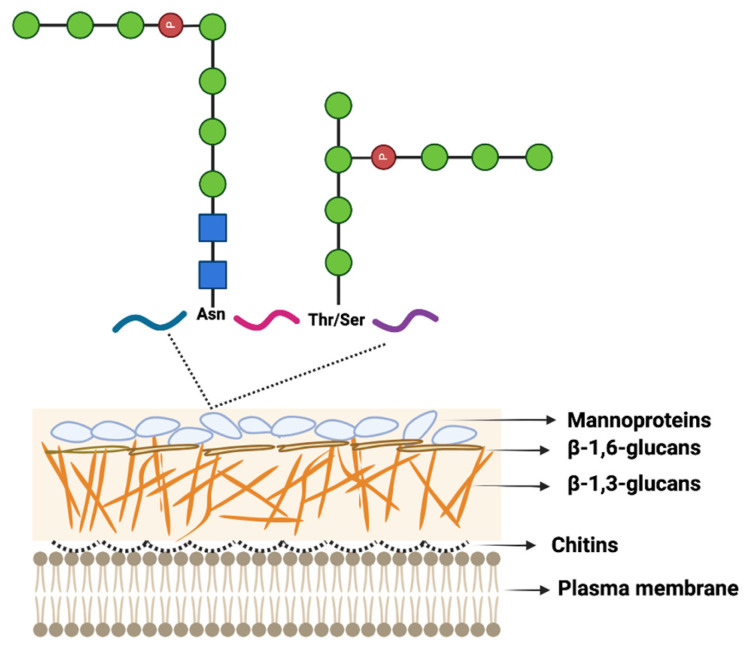
Illustration of yeast cell wall structure with mannosylated glycoproteins forming the top layer. 
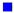
—GlcNAc, N-acetylglucosamine, 
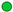
—Man, mannose, 
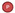
—phosphate group. The fungal cell wall is composed of chitins (1.5–6% by weight), β-linked glucans (β-1,3-glucans (30–45% by weight), β-1,6 glucan (5–10% by weight)), and mannoproteins (30–50% by weight). Chitin is a structurally important component of the fungal cell wall located closest to the plasma membrane. Branched β-1,3 glucan cross-links to chitin and is covalently linked to other polysaccharides (e.g., galactomannan and β-1,6 glucan). Mannoproteins are N- and O-glycosylated proteins. The Figure was modified from [118] and drawn with BioRender.com.

**Figure 4 jof-09-00575-f004:**
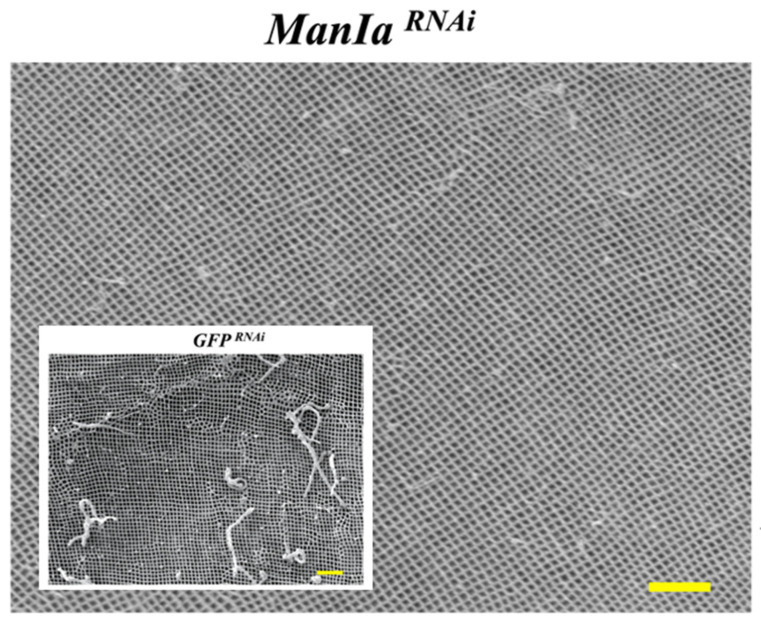
Scanning electron microscopic micrograph of the peritrophic matrix of larvae of the Colorado potato beetle (*Leptinotarsa decemlineata*) after the silencing of the ManIa gene expression (*ManIa^RNAi^*). The peritrophic matrix is a physical barrier in the gut of insects that functions to protect the midgut epithelium from mechanical damage and harm from pathogens, toxins, and other damaging chemicals. The pore width of the peritrophic matrix in *ManIa^RNAi^*-treated insects was nearly 10% smaller (83 ± 2 nm) when compared to the controls (*GFP^RNAi^*; inset micrograph; 89 ± 1 nm; significance at *p* < 0.05). Photo and data from [124]. The scale bars are 1 µm.

**Table 1 jof-09-00575-t001:** Examples of *B. bassiana*- and *M. anisopliae*-based biopesticides: Overview of the strains used, commercial names and producers, target pest insects, and region of usage. The list is a representation and is not the complete list of biopesticides in the market.

EPF Species	Commercial Name	Strain	Producer	Target Pests	Regions to Use
*B. bassiana* *	BotaniGard^TM^ ES	GHA	Laverlam	Whiteflies, aphids, weevils, mealybugs	Worldwide
	BotaniGard^TM^ MAXX				
	Mycotrol^TM^ ESO		Certis USA		USA
	Mycotrol^TM^ WPO				USA
	Aprehend^TM^		Conidia Bioscience		USA and Canada
	Broadband^TM^		Bioworks	Whiteflies, thrips, moths, stinkbugs, red spider mite, red scale	Worldwide
	BioBee^TM^	PPRI 5339	BioBee Biological Systems		Worldwide
	Bioforest^TM^	ANT-03	BioForest Technologies	Spruce budworm	USA and Canada
	BioPalm^TM^	PL11	BioPalm Manufacturing	Bagworms	Malaysia
	Boverin^TM^	ATCC 74040	Boverin Europe		Europe
	Boveril ^TM^		ESALQ Tec	Whiteflies, weevils	Brazil
	Naturalis^TM^		Koppert Biological Systems	Whiteflies, thrips, mites, aphids, tingids	Worldwide
	BioCeres^TM^	ANT-03	Bioceres	Whiteflies, aphids, thrips	Argentina, Chile, Peru, Mexico
*M. anisopliae* *	ESALQ-E9 Metarril^TM^	ESALQ-E9	EMBRAPA	Several insect and mites	Brazil
	NCIM 1311 Pacer^TM^	NCIM 1311	Chema Industries	Termites, root grub, locusts, root weevils, ants, beetles, caterpillar	Worldwide
	Bio-Blast Biological Termiticide^TM^	F52	BioProdex	Ticks, weevils, mites, thrips	USA
	Tick-Ex TM EC^TM^	ICIPE 69	Kenyan Biologics		Worldwide
	Met52 TM EC^TM^	F52	Novozymes BioAg		Worldwide
	MAS-01^TM^	Sor-1	Bio-Insumos Naturales	Coffee berry borer	USA
	Green Muscle^TM^	F52	Bioworks Inc	thrips, whiteflies, and aphids	Worldwide
	*Metarhizium* ICIPE 7^TM^	ICIPE 7	Kenya Biologics		East Africa
	*Metarhizium* AQ^TM^	CQMa421	Sinochem	Citrus root weevil	China

* The virulence of *Beauveria* and *Metarhizium* strains can be ascertained by targeted insects from a distance and the immune response is more strongly triggered by more virulent strains.

## Data Availability

Not applicable.

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
