# Peer review of "Interactions between Entomopathogenic Fungi and Insects and Prospects with Glycans"

_jof, 2023, doi:10.3390/jof9050575_

Round 1
Reviewer 1 Report
The review (Interactions between fungi and insects and related biological pest control strategies) was extensively discussed in several papers. The authors should consider the following points to improve their review.
Major points
I highly recommend that the author should discuss the role of plants in the interactions between insects and fungi.
I suggest the authors consider the Mutualistic Interactions between insects and fungi in the introduction.
The authors should consider the limitations of entomopathogenic fungi
Please provide some references to the following sentence (These include the utilization of microbial control agents against insect pests, such as bacteria, viruses and fungi).
Additionally, references are needed for the identification of Entomopathogenic fungi (EPF) line 41.
The reference Dunham Trimmer (2022) is not included in the reference list while it appears in line 68. Please also follow the journal guidelines especially in references.
I think the authors mean in line 90 Beauveria bassiana and Metarhizium anisoplia, so please correct the following sentence (In this review, we present M. anisopliae and M. anisopliae as examples).
Minor points
L21 Change to this paper provides a review of EPF
Keywords: Beauveria bassiana and Metarhizium anisoplia should be italic.
L29 Delete the purpose of
L69 Change to the biological market
L112 Delete comma after death
L 125 Please make Anisoplia austriaca in italic and add reference to this sentence.
L151 Delete the dot before (Figure 1)
L139 Grammatical error (the primary means of entry for most EPF are)
L146 Delete (through) to be read as (infect the gut wall).
L211 Change to the fungal infection.
L214 Change to mortality
L244 Grammatical error (which is secreted)
L248 Change to The binding
L261 Change to the bigger role through nodulation
L262 Change to In the later stage
L273 Change to fungal cells
L313 Change to triggering signaling pathways
L316 Change to by nearly
L325 Delete the to be read (they made the speculation)
L331 Change to strategies to increase
The English Language must be improved. The report introduced some comments, but I highly recommend that the whole review needs English Language editing.
Reviewer 2 Report
jof-2355421 Interactions between fungi and insects and related biological pest control strategies Authors reviewed the examples of B. bassiana - and M. anisopliae-based biopesticides commercially available, the mechanism of action by which EPF interact with insects, focusing on the penetration of the cuticle and the subsequent death of the host, the interactions between EPF and the insect microbiome, as well as the enhancement of the insect immune response. Finally, this review presents a potential role of N-glycans in eliciting an immune response in the insect. Article is well-written, authors summarized briefly the modern biopesticide market. However, there is lack of some important information, or there are presented very shortly. Firstly, there is little information why these two fungal species are so efficient and popular as a anti-insect agent. Secondly, there is lack important information about the defense system of insect and their role in fungal infection. Insect generally posses three barriers against infection- cuticle, the humoral and cellular immunology. Authors presented some information about the humoral defense, and few sentences about the cellular. In my opinion there is important to put more information about this topic. Also there is lack information (or very little) about cuticle, while cuticle is the most important defense mechanism against fungal infection. Minor correction, in line 90 authors used twice M. anisopliae in one sentence, probably you mean B. bassiana in one case.
Reviewer 3 Report
Dear Authors,
Your review on interactions between EPF and insect host is interesting and particularly the second part of it brings important recent information on immune response of insect and other interesting findings, e.g. on interaction with insect microbiome. However, I believe that some parts of the manuscript are not mature and require major revision. I also believe that more references should be included in review as 56 references in the present version are similar to an average research article.
Specific comments (numbers indicate lines number):
Title: should include "entomopathogenic" because some insects are also mycophagous which is also interaction but not covered in this paper
Keywords: to avoid repetition replace Entomopathogenic fungi with "mycopesticides"
27 it is not clear why it is limited to horticulture, insect pests are general problem
32 not only natural enemies but pollinators and other non-target insects, term "agricultural residue" is confusing, should it be replaced by "pesticides residue"
42 it is mostly true except species like Lecanicillium sp. which is mostly on phylloplane
43 better to write "Some of these fungi ..." because only limited number of species is utilized commercially
50 EPF collections might be better than "libraries"
52-54 this sentence should be moved after more general text, e.g. to line 60
84 should read "... the complete list of biopesticides ..."
Ingredients > EPF species
Commercial names > Commercial name
Strains > Strain
Insect control > Target pests
PFR-97 is not based on B. bassiana but on Isaria fumosorosea strain Apopka 97 (recently it was found the species was misidentified and should be Cordyceps javanica)
86-87 this table footnote seems to be out of context
90 first M. anisopliae should read B. bassiana
Page 4 - about half of the text (most of 1st and 2nd paragraph) does not fit section topic, some is repetition of already mentioned information in Introduction. This requires improvement of structure/reorganization or rewriting or modification section name or splitting it. In addition, when sentence starts with species name it is recommended to write genera name full, i.e. not abbreviated.
111 spores themself do not penetrate, it is hypha/penetration peg
125 species should be in italics
155-the end of section is also off topic
165 fungal infection
169 "... result in the spore enter inside" is not clear
171 these blastospores do not come out, it is mycelium which grows on cadaver and later produce conidia
174 first M. anisopliae should be B. bassiana
Page 8, section 3 would require more references
325 delete "the" in "so the they"
352 this is written in wrong way - first, spore addhere, then germinate, then appressorium is formed and after that fungus penetrate the cuticle
Reviewer 4 Report
These are my main comments on the MS (jof-2355421) entitled : “Interactions between fungi and insects and related biological pest control strategies”.
It is an interesting review presenting various aspects on the EPF use focusing on their interaction with insect immune system.
My proposal is to accept it for publication in your journal after major revision.
Comments
Title. The title must focus on the EPF-insect interaction. Biocontrol strategies are not described in the MS.
Introduction.
Lines 67-79. This part with statistics about biopesticides seems irrelevant. It would be better to remove it.
Figure 1. The legend does not need to be so detailed. I do not know if you can use the photos (b,c) from another study without the permission of the authors.
There are many insect species referred in the text. When a species is mentioned for the first time in the text it should be written in full : Genus species Author (Order : Family).
The subject of this review (EPF pathogenicity and interactions with immune system) has been extensively studied, especially during the last 5 years. However, authors have included only a few recent studies (only 6 references after 2019) in their relatively small reference list (56 references seems a few for a review paper). I strongly suggest checking many more references to enrich their MS. Below you can find some of them that are missing from the MS.
Li, S., Liu, F., Kang, Z., Li, X., Lu, Y., Li, Q., ... & Yin, X. (2022). Cellular immune responses of the yellow peach moth, Conogethes punctiferalis (Lepidoptera: Crambidae), to the entomopathogenic fungus, Beauveria bassiana (Hypocreales: Cordycipitaceae). Journal of Invertebrate Pathology, 194, 107826.
Mantzoukas, S., Kitsiou, F., Natsiopoulos, D., & Eliopoulos, P. A. (2022). Entomopathogenic fungi: interactions and applications. Encyclopedia, 2(2), 646-656.
Wang, J. L., Yang, K. H., Wang, S. S., Li, X. L., Liu, J., Yu, Y. X., & Liu, X. S. (2022). Infection of the entomopathogenic fungus Metarhizium rileyi suppresses cellular immunity and activates humoral antibacterial immunity of the host Spodoptera frugiperda. Pest Management Science, 78(7), 2828-2837.

I have made some suggestions for language errors on the pdf file. I feel that a more thorough language editing is needed.
Round 2
Reviewer 1 Report
The authors amended the manuscript according to my suggestions.
The authors amended the manuscript according to my suggestions.
Reviewer 2 Report
Authors responded to most of my concern. I have one obligation, authors mentioned about apresorium as important stage of fungal infection, however, there are some entomopathogens (like Conidiobolus coronatus), which do not have this stage. Please clarify this in manuscript.
Reviewer 3 Report
Dear Authors,
Thank you for revision of your manuscript. It has been improved and substantially enhanced (a lot of more references) and you addressed all my suggestions. Few places of the text, however, require additional minor corrections - please see the details below with numbers indicating lines of ms.
55 When sentence starts with species name its genera is usually written in full even it is not the first occurrence in the paper is it looks better to write Beauveria bassiana instead of B. bassiana.
65-66 Brackets seems to be redundant.
66 please delete extra "the complete biopesticides"
75 as above - full genera name is more appropriate here
87 Why do you consider "broader host range" if numbers are similar to M. anisopliae shown in text above (lines 79-80)?
143 add (Acari: Tetranychidae)
325 add author of species name
381 This sentence requires editing: "that is responsible for … (please complete), that the peritrophic"
418 Please replace "appressorium" with fungus because appressorium is the structure formed only on cuticle and by both enzymes and mechanical pressure the fungus can penetrate through.
Reviewer 4 Report
Authors have responded adequately to my suggestions. The MS can be accepted for publication.
